# Clustering from Labels and Time-Varying Graphs

**Shiau Hong Lim**
National University of Singapore
mpelsh@nus.edu.sg

**Yudong Chen**
EECS, University of California, Berkeley
yudong.chen@eecs.berkeley.edu

**Huan Xu**
National University of Singapore
mpexuh@nus.edu.sg

## Abstract

We present a general framework for graph clustering where a label is observed to each pair of nodes. This allows a very rich encoding of various types of pairwise interactions between nodes. We propose a new tractable approach to this problem based on maximum likelihood estimator and convex optimization. We analyze our algorithm under a general generative model, and provide both necessary and sufficient conditions for successful recovery of the underlying clusters. Our theoretical results cover and subsume a wide range of existing graph clustering results including planted partition, weighted clustering and partially observed graphs. Furthermore, the result is applicable to novel settings including time-varying graphs such that new insights can be gained on solving these problems. Our theoretical findings are further supported by empirical results on both synthetic and real data.

## 1 Introduction

In the standard formulation of graph clustering, we are given an unweighted graph and seek a partitioning of the nodes into disjoint groups such that members of the same group are more densely connected than those in different groups. Here, the presence of an edge represents some sort of affinity or similarity between the nodes, and the absence of an edge represents the lack thereof.

In many applications, from chemical interactions to social networks, the interactions between nodes are much richer than a simple "edge" or "non-edge". Such extra information may be used to improve the clustering quality. We may represent each type of interaction by a *label*. One simple setting of this type is weighted graphs, where instead of a 0-1 graph, we have edge weights representing the strength of the pairwise interaction. In this case the observed label between each pair is a real number. In a more general setting, the label need not be a number. For example, on social networks like Facebook, the label between two persons may be "they are friends", "they went to different schools", "they liked 21 common pages", or the concatenation of these. In such cases different labels carry different information about the underlying community structure. Standard approaches convert these pairwise interactions into a simple edge/non-edge, and then apply standard clustering algorithms, which might lose much of the information. Even in the case of a standard weighted/unweighted graph, it is not immediately clear how the graph should be used. For example, should the absence of an edge be interpreted as a *neutral* observation carrying no information, or as a *negative* observation which indicates dissimilarity between the two nodes?

We emphasize that the forms of labels can be very general. In particular, a label can take the form of a time series, i.e., the record of time varying interaction such as "A and B messaged each other on June 1st, 4th, 15th and 21st", or "they used to be friends, but they stop seeing each other since 2012". Thus, the labeled graph model is an immediate tool for analyzing time-varying graphs.

In this paper, we present a new and principled approach for graph clustering that is directly based on pairwise labels. We assume that between each pair of nodes $i$ and $j$, a label $L_{ij}$ is observed which is an element of a label set $\mathcal{L}$. The set $\mathcal{L}$ may be discrete or continuous, and need not have any structure. The standard graph model corresponds to a binary label set $\mathcal{L} = \{\text{edge}, \text{non-edge}\}$, and a weighted graph corresponds to $\mathcal{L} = \mathbb{R}$. Given the observed labels $L = (L_{ij}) \in \mathcal{L}^{n \times n}$, the goal is to partition the $n$ nodes into disjoint clusters. Our approach is based on finding a partition that optimizes a weighted objective appropriately constructed from the observed labels. This leads to a combinatorial optimization problem, and our algorithm uses its convex relaxation.

To systematically evaluate clustering performance, we consider a generalization of the stochastic block model [1] and the planted partition model [2]. Our model assumes that the observed labels are generated based on an underlying set of ground truth clusters, where pairs from the same cluster generate labels using a distribution $\mu$ over $\mathcal{L}$ and pairs from different clusters use a different distribution $\nu$. The standard stochastic block model corresponds to the case where $\mu$ and $\nu$ are two-point distributions with $\mu(\text{edge}) = p$ and $\nu(\text{edge}) = q$. We provide theoretical guarantees for our algorithm under this generalized model.

Our results cover a wide range of existing clustering settings—with equal or stronger theoretical guarantees—including the standard stochastic block model, partially observed graphs and weighted graphs. Perhaps surprisingly, our framework allows us to handle new classes of problems that are not a priori obvious to be a special case of our model, including the clustering of time-varying graphs.

## 1.1 Related work

The planted partition model/stochastic block model [1, 2] are standard models for studying graph clustering. Variants of the models cover partially observed graphs [3, 4] and weighted graphs [5, 6]. All these models are special cases of ours. Various algorithms have been proposed and analyzed under these models, such as spectral clustering [7, 8, 1], convex optimization approaches [9, 10, 11] and tensor decomposition methods [12]. Ours is based on convex optimization; we build upon and extend the approach in [13], which is designed for clustering unweighted graphs whose edges have different levels of uncertainty, a special case of our problem (cf. Section 4.2 for details).

Most related to our setting is the *labelled stochastic block model* proposed in [14] and [15]. A main difference in their model is that they assume each observation is a two-step process: first an edge/non-edge is observed; if it is an edge then a label is associated with it. In our model all observations are in the form of labels; in particular, an edge or no-edge is also a label. This covers their setting as a special case. Our model is therefore more general and natural—as a result our theory covers a broad class of subproblems including time-varying graphs. Moreover, their analysis is mainly restricted to the two-cluster setting with edge probabilities on the order of $\Theta(1/n)$, while we allow for an arbitrary number of clusters and a wide range of edge/label distributions. In addition, we consider the setting where the distributions of the labels are not precisely known. Algorithmically, they use belief propagation [14] and spectral methods [15].

Clustering time-varying graphs has been studied in various context; see [16, 17, 18, 19, 20] and the references therein. Most existing algorithms use heuristics and lack theoretical analysis. Our approach is based on a generative model and has provable performance guarantees.

## 2 Problem setup and algorithms

We assume $n$ nodes are partitioned into $r$ disjoint clusters of size at least $K$, which are unknown and considered as the ground truth. For each pair $(i, j)$ of nodes, a label $L_{ij} \in \mathcal{L}$ is observed, where $\mathcal{L}$ is the set of all possible labels.[1] These labels are generated independently across pairs according to the distributions $\mu$ and $\nu$. In particular, the probability of observing the label $L_{ij}$ is $\mu(L_{ij})$ if $i$ and $j$ are in the same cluster, and $\nu(L_{ij})$ otherwise. The goal is to recover the ground truth clusters given the labels. Let $L = (L_{ij}) \in \mathcal{L}^{n \times n}$ be the matrix of observed labels. We represent the true clusters by an $n \times n$ *cluster matrix* $Y^*$, where $Y^*_{ij} = 1$ if nodes $i$ and $j$ belong to the same cluster and $Y^*_{ij} = 0$ otherwise (we use the convention $Y^*_{ii} = 1$ for all $i$). The problem is therefore to find $Y^*$ given $L$.

We take an optimization approach to this problem. To motivate our algorithm, first consider the case of clustering a weighted graph, where all labels are real numbers. Positive weights indicate in-cluster interaction while negative weights indicate cross-cluster interaction. A natural approach is to cluster the nodes in a way that maximizes the total weight inside the clusters (this is equivalent to *correlation clustering* [21]). Mathematically, this is to find a clustering, represented by a cluster matrix $Y$, such that $\sum_{i,j} L_{ij} Y_{ij}$ is maximized. For the case of general labels, we pick a *weight function* $w : \mathcal{L} \mapsto \mathbb{R}$, which assigns a number $W_{ij} = w(L_{ij})$ to each label, and then solve the following max-weight problem:

$$\max_Y \ \langle W, Y \rangle \quad \text{s.t. } Y \text{ is a cluster matrix;} \tag{1}$$

here $\langle W, Y \rangle := \sum_{ij} W_{ij} Y_{ij}$ is the standard trace inner product. Note that this effectively converts the problem of clustering from labels into a weighted clustering problem.

The program (1) is non-convex due to the constraint. Our algorithm is based on a convex relaxation of (1), using the now well-known fact that a cluster matrix is a block-diagonal 0-1 matrix and thus has nuclear norm[2] equal to $n$ [22, 3, 23]. This leads to the following convex optimization problem:

$$\max_Y \quad \langle W, Y \rangle \tag{2}$$
$$\text{s.t.} \quad \|Y\|_* \leq n; \ 0 \leq Y_{ij} \leq 1, \forall (i,j).$$

We say that this program succeeds if it has a unique optimal solution equal to the true cluster matrix $Y^*$. We note that a related approach is considered in [13], which is discussed in section 4.

One has the freedom of choosing the weight function $w$. Intuitively, $w$ should assign $w(L_{ij}) > 0$ to a label $L_{ij}$ with $\mu(L_{ij}) > \nu(L_{ij})$, so the program (2) is encouraged to place $i$ and $j$ in the same cluster, the more likely possibility; similarly we should have $w(L_{ij}) < 0$ if $\mu(L_{ij}) < \nu(L_{ij})$. A good weight function should further reflect the information in $\mu$ and $\nu$. Our theoretical results in section 3 characterize the performance of the program (2) for *any* given weight function; building on this, we further derive the optimal choice for the weight function.

## 3 Theoretical results

In this section, we provide theoretical analysis for the performance of the convex program (2) under the probabilistic model described in section 2. The proofs are given in the supplementary materials.

Our main result is a general theorem that gives sufficient conditions for (2) to recover the true cluster matrix $Y^*$. The conditions are stated in terms of the label distribution $\mu$ and $\nu$, the minimum size of the true clusters $K$, and any given weight function $w$. Define $\mathbb{E}_\mu w := \sum_{l \in \mathcal{L}} w(l)\mu(l)$ and $\text{Var}_\mu w := \sum_{l \in \mathcal{L}} [w(l) - \mathbb{E}_\mu w]^2 \mu(l)$; $\mathbb{E}_\nu w$ and $\text{Var}_\nu w$ are defined similarly.

**Theorem 1** (Main). *Suppose $b$ is any number that satisfies $|w(l)| \leq b, \forall l \in \mathcal{L}$ almost surely. There exists a universal constant $c > 0$ such that if*

$$-\mathbb{E}_\nu w \geq c \frac{b \log n + \sqrt{K \log n}\sqrt{\text{Var}_\nu w}}{K}, \tag{3}$$

$$\mathbb{E}_\mu w \geq c \frac{b \log n + \sqrt{n \log n}\sqrt{\max(\text{Var}_\mu w, \text{Var}_\nu w)}}{K}, \tag{4}$$

*then $Y^*$ is the unique solution to (2) with probability at least $1 - n^{-10}$.* [3]

The theorem holds for any given weight function $w$. In the next two subsections, we show how to choose $w$ optimally, and then address the case where $w$ deviates from the optimal choice.

### 3.1 Optimal weights

A good candidate for the weight function $w$ can be derived from the maximum likelihood estimator (MLE) of $Y^*$. Given the observed labels $L$, the log-likelihood of the true cluster matrix taking

the value $Y$ is

$$\log \Pr(L|Y^* = Y) = \sum_{i,j} \log \left[ \mu(L_{ij})^{Y_{ij}} \nu(L_{ij})^{1-Y_{ij}} \right] = \langle W, Y \rangle + c$$

where $c$ is independent of $Y$ and $W$ is given by the weight function $w(l) = w^{\text{MLE}}(l) := \log \frac{\mu(l)}{\nu(l)}$.
The MLE thus corresponds to using the log-likelihood ratio $w^{\text{MLE}}(\cdot)$ as the weight function. The
following theorem is a consequence of Theorem 1 and characterizes the performance of using the
MLE weights. In the sequel, we use $D(\cdot\|\cdot)$ to denote the KL divergence between two distributions.

**Theorem 2** (MLE). *Suppose $w^{MLE}$ is used, and $b$ and $\zeta$ are any numbers which satisfy with*
$D(\nu\|\mu) \le \zeta D(\mu\|\nu)$ *and* $\left| \log \frac{\mu(l)}{\nu(l)} \right| \le b, \forall l \in \mathcal{L}$. *There exists a universal constant $c > 0$ such*
*that $Y^*$ is the unique solution to (2) with probability at least $1 - n^{-10}$ if*

$$D(\nu\|\mu) \ge c(b+2)\frac{\log n}{K}, \tag{5}$$

$$D(\mu\|\nu) \ge c(\zeta+1)(b+2)\left(\frac{n \log n}{K^2}\right). \tag{6}$$

*Moreover, we always have $D(\nu\|\mu) \le (2b+3)D(\mu\|\nu)$, so we can take $\zeta = 2b+3$.*

Note that the theorem has the intuitive interpretation that the in/cross-cluster label distributions $\mu$ and
$\nu$ should be sufficiently different, measured by their KL divergence. Using a classical result in infor-
mation theory [24], we may replace the KL divergences with a quantity that is often easier to work
with, as summarized below. The LHS of (7) is sometimes called the triangle discrimination [24].

**Corollary 1** (MLE 2). *Suppose $w^{MLE}$ is used, and $b$, $\zeta$ are defined as in Theorem 2. There exists a*
*universal constant $c$ such that $Y^*$ is the unique solution to (2) with probability at least $1 - n^{-10}$ if*

$$\sum_{l \in \mathcal{L}} \frac{(\mu(l) - \nu(l))^2}{\mu(l) + \nu(l)} \ge c(\zeta+1)(b+2)\left(\frac{n \log n}{K^2}\right). \tag{7}$$

*We may take $\zeta = 2b+3$.*

The MLE weight $w^{\text{MLE}}$ turns out to be near-optimal, at least in the two-cluster case, in the sense that
no other weight function (in fact, no other algorithm) has significantly better performance. This is
shown by establishing a necessary condition for *any* algorithm to recover $Y^*$. Here, an algorithm is
a measurable function $\hat{Y}$ that maps the data $L$ to a clustering (represented by a cluster matrix).

**Theorem 3** (Converse). *The following holds for some universal constants $c, c' > 0$. Suppose $K = \frac{n}{2}$, and $b$ defined in Theorem 2 satisfies $b \le c'$. If*

$$\sum_{l \in \mathcal{L}} \frac{(\mu(l) - \nu(l))^2}{\mu(l) + \nu(l)} \le \frac{c \log n}{n}, \tag{8}$$

*then $\inf_{\hat{Y}} \sup_{Y^*} \mathbb{P}(\hat{Y} \ne Y^*) \ge \frac{1}{2}$, where the supremum is over all possible cluster matrices.*

Under the assumption of Theorem 3, the conditions (7) and (8) match up to a constant factor.

**Remark.** *The MLE weight $|w^{MLE}(l)|$ becomes large if $\mu(l) = o(\nu(l))$ or $\nu(l) = o(\mu(l))$, i.e., when*
*the in-cluster probability is negligible compared to the cross-cluster one (or the other way around).*
*It can be shown that in this case the MLE weight is actually order-wise better than a bounded weight*
*function. We give this result in the supplementary material due to space constraints.*

## 3.2 Monotonicity

We sometimes do not know the exact true distributions $\mu$ and $\nu$ to compute $w^{\text{MLE}}$. Instead, we might
compute the weight using the log likelihood ratios of some "incorrect" distribution $\bar{\mu}$ and $\bar{\nu}$. Our
algorithm has a nice *monotonicity* property: as long as the divergence of the true $\mu$ and $\nu$ is larger
than that of $\bar{\mu}$ and $\bar{\nu}$ (hence an "easier" problem), then the problem should still have the same, if not
better probability of success, even though the wrong weights are used.

We say that $(\mu, \nu)$ is *more divergent* then $(\bar{\mu}, \bar{\nu})$ if, for each $l \in \mathcal{L}$, we have that either

$$\frac{\mu(l)}{\nu(l)} \ge \frac{\mu(l)}{\bar{\nu}(l)} \ge \frac{\bar{\mu}(l)}{\bar{\nu}(l)} \ge 1 \quad \text{or} \quad \frac{\nu(l)}{\mu(l)} \ge \frac{\nu(l)}{\bar{\mu}(l)} \ge \frac{\bar{\nu}(l)}{\bar{\mu}(l)} \ge 1.$$

**Theorem 4** (Monotonicity)**.** *Suppose we use the weight function $w(l) = \log \frac{\bar{\mu}(l)}{\bar{\nu}(l)}, \forall l$, while the actual label distributions are $\mu$ and $\nu$. If the conditions in Theorem 2 or Corollary 1 hold with $\mu, \nu$ replaced by $\bar{\mu}, \bar{\nu}$, and $(\mu, \nu)$ is more divergent than $(\bar{\mu}, \bar{\nu})$, then with probability at least $1 - n^{-10}$ $Y^*$ is the unique solution to (2).*

This result suggests that one way to choose the weight function is by using the log-likelihood ratio based on a "conservative" estimate (i.e., a less divergent one) of the true label distribution pair.

### 3.3 Using inaccurate weights

In the previous subsection we consider using a conservative log-likelihood ratio as the weight. We now consider a more general weight function $w$ which need not be conservative, but is only required to be not too far from the true log-likelihood ratio $w^{\text{MLE}}$. Let

$$\varepsilon(l) := w(l) - w^{\text{MLE}}(l) = w(l) - \log \frac{\mu(l)}{\nu(l)}$$

be the error for each label $l \in \mathcal{L}$. Accordingly, let $\Delta_\mu := \sum_{l \in \mathcal{L}} \mu(l)\varepsilon(l)$ and $\Delta_\nu := \sum_{l \in \mathcal{L}} \nu(l)\varepsilon(l)$ be the average errors with respect to $\mu$ and $\nu$. Note that $\Delta_\mu$ and $\Delta_\nu$ can be either positive or negative. The following characterizes the performance of using such a $w$.

**Theorem 5** (Inaccurate Weights)**.** *Let $b$ and $\zeta$ be defined as in Theorem 2. If the weight $w$ satisfies*

$$|w(l)| \leq \lambda \left| \log \frac{\mu(l)}{\nu(l)} \right|, \forall l \in \mathcal{L}, \quad |\Delta_\mu| \leq \gamma D(\mu\|\nu), \quad |\Delta_\nu| \leq \gamma D(\nu\|\mu)$$

*for some $\gamma < 1$ and $\lambda > 0$. Then $Y^*$ is unique solution to (2) with probability at least $1 - n^{-10}$ if*

$$D(\nu\|\mu) \geq c\frac{\lambda^2}{(1-\gamma)^2}(b+2)\frac{\log n}{K} \quad \text{and} \quad D(\mu\|\nu) \geq c\frac{\lambda^2}{(1-\gamma)^2}(\zeta+1)(b+2)\left(\frac{n \log n}{K^2}\right).$$

Therefore, as long as the errors $\Delta_\mu$ and $\Delta_\nu$ in $w$ are not too large, the condition for recovery will be order-wise similar to that in Theorem 2 for using the MLE weight. The numbers $\lambda$ and $\gamma$ measure the amount of inaccuracy in $w$ w.r.t. $w^{\text{MLE}}$. The last two conditions in Theorem 5 thus quantify the relation between the inaccuracy in $w$ and the price we need to pay for using such a weight.

## 4 Consequences and applications

We apply the general results in the last section to different special cases. In sections 4.1 and 4.2, we consider two simple settings and show that two immediate corollaries of our main theorems recover, and in fact improve upon, existing results. In sections 4.3 and 4.4, we turn to the more complicated setting of clustering time-varying graphs and derive several novel results.

### 4.1 Clustering a Gaussian matrix with partial observations

Analogous to the planted partition model for unweighted graphs, the bi-clustering [5] or submatrix-localization [6, 23] problem concerns with weighted graph whose adjacency matrix has Gaussian entries. We consider a generalization of this problem where some of the entries are unobserved.

Specifically, we observe a matrix $L \in (\mathbb{R} \cup \{?\})^{n \times n}$, which has $r$ submatrices of size $K \times K$ with disjoint row and column support, such that $L_{ij} = ?$ (meaning unobserved) with probability $1 - s$ and otherwise $L_{ij} \sim \mathcal{N}(u_{ij}, 1)$. Here the means of the Gaussians satisfy: $u_{ij} = \bar{u}$ if $(i, j)$ is inside the submatrices and $u_{ij} = \underline{u}$ if outside, where $\bar{u} > \underline{u} \geq 0$. Clustering is equivalent to locating these submatrices with elevated mean, given the large Gaussian matrix $L$ with partial observations.[4]

This is a special case of our labeled framework with $\mathcal{L} = \mathbb{R} \cup \{?\}$. Computing the log-likelihood ratios for two Gaussians, we obtain $w^{\text{MLE}}(L_{ij}) = 0$ if $L_{ij} = ?$, and $w^{\text{MLE}}(L_{ij}) \propto L_{ij} - (\bar{u} + \underline{u})/2$ otherwise. This problem is interesting only when $\bar{u} - \underline{u} \lesssim \sqrt{\log n}$ (otherwise simple element-wise thresholding [5, 6] finds the submatrices), which we assume to hold. Clearly $D(\mu\|\nu) = D(\nu\|\mu) = \frac{1}{4}s(\bar{u} - \underline{u})^2$. The following can be proved using our main theorems (proof in the appendix).

**Corollary 2** (Gaussian Graphs). *Under the above setting, $Y^*$ is the unique solution to (2) with weights $w = w^{MLE}$ with probability at least $1 - 2n^{-10}$ provided*

$$s\left(\bar{u} - \underline{u}\right)^2 \geq c\frac{n\log^3 n}{K^2}.$$

In the fully observed case, this recovers the results in [23, 5, 6] up to log factors. Our results are more general as we allow for partial observations, which is not considered in previous work.

### 4.2 Planted Partition with non-uniform edge densities

The work in [13] considers a variant of the planted partition model with non-uniform edge densities, where each pair $(i, j)$ has an edge with probability $1 - u_{ij} > 1/2$ if they are in the same cluster, and with probability $u_{ij} < 1/2$ otherwise. The number $u_{ij}$ can be considered as a measure of the level of uncertainty in the observation between $i$ and $j$, and is known or can be estimated in applications like cloud-clustering. They show that using the knowledge of $\{u_{ij}\}$ improves clustering performance, and such a setting covers clustering of partially observed graphs that is considered in [11, 3, 4].

Here we consider a more general setting that does not require the in/cross-cluster edge density to be symmetric around $\frac{1}{2}$. Suppose each pair $(i, j)$ is associated with two numbers $p_{ij}$ and $q_{ij}$, such that if $i$ and $j$ are in the same cluster (different clusters, resp.), then there is an edge with probability $p_{ij}$ ($q_{ij}$, resp.); we know $p_{ij}$ and $q_{ij}$ but not which of them is the probability that generates the edge. The values of $p_{ij}$ and $q_{ij}$ are generated i.i.d. randomly as $(p_{ij}, q_{ij}) \sim \mathbb{D}$ by some distribution $\mathbb{D}$ on $[0, 1] \times [0, 1]$. The goal is to find the clusters given the graph adjacency matrix $A$, $(p_{ij})$ and $(q_{ij})$.

This model is a special case of our labeled framework. The labels have the form $L_{ij} = (A_{ij}, p_{ij}, q_{ij}) \in \mathcal{L} = \{0, 1\} \times [0, 1] \times [0, 1]$, generated by the distributions

$$\mu(l) = \begin{cases} p\mathbb{D}(p, q), & l = (1, p, q) \\ (1 - p)\mathbb{D}(p, q), & l = (0, p, q) \end{cases} \qquad \nu(l) = \begin{cases} q\mathbb{D}(p, q), & l = (1, p, q) \\ (1 - q)\mathbb{D}(p, q), & l = (0, p, q). \end{cases}$$

The MLE weight has the form $w^{\text{MLE}}(L_{ij}) = A_{ij} \log \frac{p_{ij}}{q_{ij}} + (1 - A_{ij}) \log \frac{1 - p_{ij}}{1 - q_{ij}}$. It turns out it is more convenient to use a conservative weight in which we replace $p_{ij}$ and $q_{ij}$ with $\bar{p}_{ij} = \frac{3}{4}p_{ij} + \frac{1}{4}q_{ij}$ and $\bar{q}_{ij} = \frac{1}{4}p_{ij} + \frac{3}{4}q_{ij}$. Applying Theorem 4 and Corollary 1, we immediately obtain the following.

**Corollary 3** (Non-uniform Density). *Program (2) recovers $Y^*$ with probability at least $1 - n^{-10}$ if*

$$\mathbb{E}_{\mathbb{D}}\left[\frac{(p_{ij} - q_{ij})^2}{p_{ij}(1 - q_{ij})}\right] \geq c\frac{n\log n}{K^2}, \forall (i.j).$$

*Here $\mathbb{E}_{\mathbb{D}}$ is the expectation w.r.t. the distribution $\mathbb{D}$, and LHS above is in fact independent of $(i, j)$.*

Corollary 3 improves upon existing results for several settings.

- **Clustering partially observed graphs.** Suppose $\mathbb{D}$ is such that $p_{ij} = p$ and $q_{ij} = q$ with probability $s$, and $p_{ij} = q_{ij}$ otherwise, where $p > q$. This extends the standard planted partition model: each pair is unobserved with probability $1 - s$. For this setting we require

$$\frac{s(p - q)^2}{p(1 - q)} \gtrsim \frac{n\log n}{K^2}.$$

  When $s = 1$. this matches the best existing bounds for standard planted partition [9, 12] up to a log factor. For the partial observation setting with $s \leq 1$, the work in [4] gives a similar bound under the additional assumption $p > 0.5 > q$, which is not required by our result. For general $p$ and $q$, the best existing bound is given in [3, 9], which replaces unobserved entries with 0 and requires the condition $\frac{s(p-q)^2}{p(1-sq)} \gtrsim \frac{n\log n}{K^2}$. Our result is tighter when $p$ and $q$ are close to 1.

- **Planted partition with non-uniformity.** The model and algorithm in [13] is a special case of ours with symmetric densities $p_{ij} \equiv 1 - q_{ij}$, for which we recover their result $\mathbb{E}_{\mathbb{D}}\left[(1 - 2q_{ij})^2\right] \gtrsim \frac{n\log n}{K^2}$. Corollary 3 is more general as it removes the symmetry assumption.

## 4.3 Clustering time-varying multiple-snapshot graphs

Standard graph clustering concerns with clustering on a single, static graph. We now consider a setting where the graph can be time-varying. Specifically, we assume that for each time interval $t = 1, 2, \ldots, T$, we observed a snapshot of the graph $L^{(t)} \in \mathcal{L}^{n \times n}$. We assume each snapshot is generated by the distributions $\mu$ and $\nu$, independent of other snapshots.

We can map this problem into our original labeled framework, by considering the whole time sequence of $\bar{L}_{ij} := (L_{ij}^{(1)}, \ldots, L_{ij}^{(T)})$ observed at the pair $(i, j)$ as a single label. In this case the label set is thus the set of all possible sequences, i.e., $\bar{\mathcal{L}} = (\mathcal{L})^T$, and the label distributions are (with a slight abuse of notation) $\mu(\bar{L}_{ij}) = \mu(L_{ij}^{(1)}) \ldots \mu(L_{ij}^{(T)})$, with $\nu(\cdot)$ given similarly. The MLE weight (normalized by $T$) is thus the average log-likelihood ratio:

$$w^{\text{MLE}}(\bar{L}_{ij}) = \frac{1}{T} \log \frac{\mu(L_{ij}^{(1)}) \ldots \mu(L_{ij}^{(T)})}{\nu(L_{ij}^{(1)}) \ldots \nu(L_{ij}^{(T)})} = \frac{1}{T} \sum_{t=1}^{T} \log \frac{\mu(L_{ij}^{(t)})}{\nu(L_{ij}^{(t)})}.$$

Since $w^{\text{MLE}}(\bar{L}_{ij})$ is the average of $T$ independent random variables, its variance scales with $\frac{1}{T}$. Applying Theorem 1, with almost identical proof as in Theorem 2 we obtain the following:

**Corollary 4** (Independent Snapshots). *Suppose* $|\log \frac{\mu(l)}{\nu(l)}| \leq b, \forall l \in \mathcal{L}$ *and* $D(\nu||\mu) \leq \zeta D(\mu||\nu)$. *The program (2) with MLE weights given recovers* $Y^*$ *with probability at least* $1 - n^{-10}$ *provided*

$$D(\nu||\mu) \geq c(b+2)\frac{\log n}{K}, \tag{9}$$

$$D(\mu||\nu) \geq c(b+2) \max \left\{ \frac{\log n}{K}, (\zeta+1)\frac{n \log n}{TK^2} \right\}. \tag{10}$$

Setting $T = 1$ recovers Theorem 2. When the second term in (10) dominates, the corollary says that the problem becomes easier if we observe more snapshots, with the tradeoff quantified precisely.

## 4.4 Markov sequence of snapshots

We now consider the more general and useful setting where the snapshots form a Markov chain. For simplicity we assume that the Markov chain is time-invariant and has a unique stationary distribution which is also the initial distribution. Therefore, the observations $L_{ij}^{(t)}$ at each $(i, j)$ are generated by first drawing a label from the stationary distribution $\bar{\mu}$ (or $\bar{\nu}$) at $t = 1$, then applying a one-step transition to obtain the label at each subsequent $t$. In particular, given the previously observed label $l$, let the intra-cluster and inter-cluster conditional distributions be $\mu(\cdot|l)$ and $\nu(\cdot|l)$. We assume that the Markov chains with respect to both $\mu$ and $\nu$ are geometrically ergodic such that for any $\tau \geq 1$, and label-pair $L^{(1)}, L^{(\tau+1)}$,

$$|\Pr_\mu(L^{(\tau+1)}|L^{(1)}) - \bar{\mu}(L^{(\tau+1)})| \leq \kappa \gamma^\tau \quad \text{and} \quad |\Pr_\nu(L^{(\tau+1)}|L^{(1)}) - \bar{\nu}(L^{(\tau+1)})| \leq \kappa \gamma^\tau$$

for some constants $\kappa \geq 1$ and $\gamma < 1$ that only depend on $\mu$ and $\nu$. Let $D_l(\mu||\nu)$ be the KL-divergence between $\mu(\cdot|l)$ and $\nu(\cdot|l)$; $D_l(\nu||\mu)$ is similarly defined. Let $\mathbb{E}_{\bar{\mu}} D_l(\mu||\nu) = \sum_{l \in \mathcal{L}} \bar{\mu}(l) D_l(\mu||\nu)$ and similarly for $\mathbb{E}_{\bar{\nu}} D_l(\nu||\mu)$. As in the previous subsection, we use the average log-likelihood ratio as the weight. Define $\lambda = \frac{\kappa}{(1-\gamma)\min_l\{\bar{\mu}(l),\bar{\nu}(l)\}}$. Applying Theorem 1 gives the following corollary. See sections H–I in the supplementary material for the proof and additional discussion.

**Corollary 5** (Markov Snapshots). *Under the above setting, suppose for each label-pair* $(l, l')$, $\left|\log \frac{\bar{\mu}(l)}{\bar{\nu}(l)}\right| \leq b$, $\left|\log \frac{\mu(l'|l)}{\nu(l'|l)}\right| \leq b$, $D(\bar{\nu}||\bar{\mu}) \leq \zeta D(\bar{\mu}||\bar{\nu})$ *and* $\mathbb{E}_{\bar{\nu}} D_l(\nu||\mu) \leq \zeta \mathbb{E}_{\bar{\mu}} D_l(\mu||\nu)$. *The program (2) with MLE weights recovers* $Y^*$ *with probability at least* $1 - n^{-10}$ *provided*

$$\frac{1}{T} D(\bar{\nu}||\bar{\mu}) + \left(1 - \frac{1}{T}\right) \mathbb{E}_{\bar{\nu}} D_l(\nu||\mu) \geq c(b+2)\frac{\log n}{K} \tag{11}$$

$$\frac{1}{T} D(\bar{\mu}||\bar{\nu}) + \left(1 - \frac{1}{T}\right) \mathbb{E}_{\bar{\mu}} D_l(\mu||\nu) \geq c(b+2) \max \left\{ \frac{\log n}{K}, (\zeta+1)\lambda\frac{n \log n}{TK^2} \right\}. \tag{12}$$

As an illuminating example, consider the case where $\bar{\mu} \approx \bar{\nu}$, i.e., the marginal distributions for individual snapshots are identical or very close. It means that the information is contained in the *change* of labels, but not in the individual labels, as made evident in the LHSs of (11) and (12). In this case, *it is necessary to use the temporal information in order to perform clustering*. Such information would be lost if we disregard the ordering of the snapshots, for example, by aggregating or averaging the snapshots then apply a single-snapshot clustering algorithm. This highlights an essential difference between clustering time-varying graphs and static graphs.

## 5  Empirical results

To solve the convex program (2), we follow [13, 9] and adapt the ADMM algorithm by [25]. We perform 100 trials for each experiment, and report the success rate, i.e., the fraction of trials where the ground-truth clustering is fully recovered. Error bars show $95\%$ confidence interval. Additional empirical results are provided in the supplementary material.

We first test the planted partition model with partial observations under the challenging sparse ($p$ and $q$ close to 0) and dense settings ($p$ and $q$ close to 1); cf. section 4.2. Figures 1 and 2 show the results for $n = 1000$ with 4 equal-size clusters. In both cases, each pair is observed with probability 0.5. For comparison, we include results for the MLE weights as well as the linear weights (based on linear approximation of the log-likelihood ratio), uniform weights and a imputation scheme where all unobserved entries are assumed to be "no-edge".

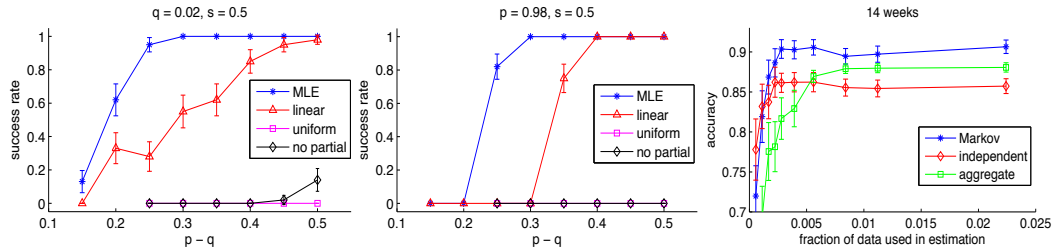

Figure 1: Sparse graphs    Figure 2: Dense graphs    Figure 3: Reality Mining dataset

Corollary 3 predicts more success as the ratio $\frac{s(p-q)^2}{p(1-q)}$ gets larger. All else being the same, distributions with small $\zeta$ (sparse) are "easier" to solve. Both predictions are consistent with the empirical results in Figs. 1 and 2. The results also show that the MLE weights outperform the other weights.

For real data, we use the Reality Mining dataset [26], which contains individuals from two main groups, the MIT Media Lab and the Sloan Business School, which we use as the ground-truth clusters. The dataset records when two individuals interact, i.e., become proximal of each other or make a phone call, over a 9-month period. We choose a window of 14 weeks (the Fall semester) where most individuals have non-empty interaction data. These consist of 85 individuals with 25 of them from Sloan. We represent the data as a time-varying graph with 14 snapshots (one per week) and two labels—an "edge" if a pair of individuals interact within the week, and "no-edge" otherwise.

We compare three models: Markov sequence, independent snapshots, and the aggregate (union) graphs. In each trial, the in/cross-cluster distributions are estimated from a fraction of randomly selected pairwise interaction data. The vertical axis in Figure 3 shows the fraction of pairs whose cluster relationship are correctly identified. From the results, we infer that the interactions between individuals are likely not independent across time, and are better captured by the Markov model.

**Acknowledgments**

S.H. Lim and H. Xu were supported by the Ministry of Education of Singapore through AcRF Tier Two grant R-265-000-443-112. Y. Chen was supported by NSF grant CIF-31712-23800 and ONR MURI grant N00014-11-1-0688.

## Footnotes

[1]Note that $\mathcal{L}$ does not have to be finite. Although some of the results are presented for finite $\mathcal{L}$, they can be easily adapted to the other cases, for instance, by replacing summation with integration.

[2]The nuclear norm of a matrix is defined as the sum of its singular values. A cluster matrix is positive semidefinite so its nuclear norm is equal to its trace.

[3]In all our results, the choice $n^{-10}$ is arbitrary. In particular, the constant $c$ scales linearly with the exponent.

[4]Here for simplicity we consider the clustering setting instead of bi-clustering. The latter setting corresponds to rectangular $L$ and submatrices. Extending our results to this setting is relatively straightforward.

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
