[Supplementary Material · final_supp.pdf]

# Appendices

In the first section, we present additional empirical results. The rest of this appendix contains proofs of the theoretical results in the main text.

## A    Additional experiment results

In this section we provide more experimental results.

### A.1    Clustering with general labels

In this subsection, we evaluate graph clustering performance on a generic graph model with 5 labels. We use $n = 200$ with 4 equal-size clusters. In each experiment, two distributions $\mu$ and $\nu$ are randomly chosen as the in-cluster and the cross-cluster distributions. Then, 100 random graphs are generated using this $(\mu, \nu)$ pair and each clustering outcome is checked against the corresponding ground truth. This is repeated 500 times to get a large variety of $(\mu, \nu)$ pairs. Based on Theorem 2, the KL-divergence between $\mu$ and $\nu$ is the key deciding factor for the successful recovery of the underlying true clusters. The results are shown in Figure 4. We use the sum of $D(\mu||\nu)$ and $D(\nu||\mu)$ as the predictor (the horizontal axis is cut off after 2 since beyond this range all success rates are essentially 1). The results indeed support the theoretical prediction.

Figure 4: Clustering with 5 labels

### A.2    Clustering sparse/dense graphs with partial observations

Here, we repeat the experiments described in Section 5 for a smaller $n = 200$ and show the analogue of Figs. 1 and 2 in Figs. 5 and 6. Clustering is more difficult for smaller $n$ and therefore we increase the observation rate from 0.5 to 0.8 (to show a more interesting region). Qualitatively the results are very similar. As predicted, the dense case is more challenging, and the "no partial" approach does not perform well in this case.

We next examine the effect of varying cluster sizes on the performance, with a fixed total number of nodes. Figures 7 and 8 show clustering results with various cluster sizes $(K)$, for $n = 400$. We choose a particular $p$ and $q$ that shows an interesting region. As expected, the success rates improve when $K$ grows. Qualitatively the results remain similar for other $p$ and $q$.

Figure 5: Clustering sparse graph

Figure 6: Clustering dense graph

Figure 7: Clustering sparse graph

Figure 8: Clustering dense graph

## A.3 Time-varying graphs

Figure 9: Independent snapshots

Figure 10: Markov label sequence

We next test clustering performance on time-varying graphs. Figure 9 shows results for clustering based on multiple independent snapshots of labeled graphs. Each graph is generated according to the planted partition model with partial observation and a fixed error rate. We tested a wide range of error rates and the horizontal axis tracks the corresponding KL divergence between $\mu$ and $\nu$. Figure 9 shows the results for 200 nodes with 8 equal-size clusters. As predicted by Corollary 4, the clustering performance improves with the number of snapshots ($T$).

Figure 10 shows results for Markov label sequence. Here, we test a very simple model with two labels {interaction, no-interaction}. For within-cluster pairs, the probability of interaction is greater ($0.5 + \epsilon$) in the next time-step if there is no interaction in the current time-step, and vice versa.

For inter-cluster pairs, the probability of interaction/no-interaction is completely random. This is the case where both the marginal and the stationary distributions for $\mu$ and $\nu$ are identical in every time-step, and therefore at least two consecutive observations are needed for informative clustering. The figure shows results for 200 nodes with 8 equal-size clusters, with the horizontal axis tracking the average KL-divergence between the conditional distributions (by varying $\epsilon$). As predicted by Corollary 5, the performance improves with $T$.

## A.4 Reality Mining dataset

In this subsection we show additional results on the Reality Mining dataset. Figure 11 shows the analogue of Fig. 3 but only for a period of 5 weeks (i.e. 5 snapshots). In this case, we see that all three models have worse in/cross-cluster accuracy. In particular, the independent model actually performs better than simply using the aggregate graph.

Figure 11: Reality Mining dataset, shorter period

In the above experiment, although the in-cluster and off-cluster distributions are estimated from only a tiny fraction of the data, the "training" data is still part of the test data. We now test a second scheme. In this scheme, a small number of subjects are randomly chosen in each trial and the inter-action data between these chosen set and the other subjects are used for parameter estimation. These subjects are then removed from the total set, and graph clustering is performed on the remaining subjects. Figure 12 shows the results for both 5-week and 14-week. Qualitatively, the results are similar to those from Figs. 3 and 11.

Figure 12: Reality Mining dataset, scheme 2

# B  A general theorem and high confidence observations

In this section, we state and prove a general theorem which implies the main Theorem 1 in the main text. A benefit of considering such a general theorem is that we can cover the case where some of the observations are highly certain, or are given as hard constraints. Naturally, the weights assigned to these observations would have arbitrarily large magnitudes.

Let
$$S_b = \{l \in \mathcal{L} : |w(l)| \geq b\}$$
and $s_{\mu,b} = \sum_{l \in S_b} \mu(l)$.

Let
$$\tilde{w}_b(l) = \begin{cases} w(l) & \text{if } l \notin S_b, \\ 0 & \text{otherwise} \end{cases}$$

The following theorem, which generalizes Theorem 1, gives the sufficient conditions for successful recovery:

**Theorem 6.** *Suppose there exists $b$ such that all entries with $|W_{ij}| \geq b$ are consistent, i.e.:*
$$W_{ij}(2Y_{ij}^* - 1) > 0 \tag{13}$$

*and $K$ is the minimum cluster size. Let $b' = \max_l |\tilde{w}_b(l)|$. If the following holds:*
$$-\mathbb{E}_\nu \tilde{w}_b \geq c \frac{b'\beta \log n + \sqrt{K\beta \log n}\sqrt{\text{Var}_\nu \tilde{w}_b}}{K} \tag{14}$$

*and at least one of the following holds:*

*1.*
$$\mathbb{E}_\mu \tilde{w}_b > c \frac{b'\beta \log n + \sqrt{n\beta \log n}\sqrt{\max(\text{Var}_\mu \tilde{w}_b, \text{Var}_\nu \tilde{w}_b)}}{K} \tag{15}$$

*2.*
$$s_{\mu,b} + \frac{\mathbb{E}_\mu \tilde{w}_b}{b} > c \max\left(\frac{\beta \log n}{K}, s_{\mu,b}(1 - s_{\mu,b}), \frac{n \max(\text{Var}_\mu \tilde{w}_b, \text{Var}_\nu \tilde{w}_b)}{Kb^2}\right) \tag{16}$$

*then with probability at least $1 - n^{-\beta}$ $Y^*$ is the unique solution to (2).*

It is easy to see that if $w$ is bounded, then we can choose $b$ such that $S_b$ is an empty set. In this case, condition (16) can be ignored and we have Theorem 1. On the other hand, if there exist hard constraints with $b \to \infty$, then (16) reduces to
$$s_{\mu,b} > c \frac{\beta \log n}{K}.$$

This is significant since in this case, the overall condition for successful recovery only depends logarithmically on $n$.

## B.1  Proof of Theorem 6

We now prove the generalized Theorem 6.

### B.1.1  Preliminaries

Let $\Omega = \{(i,j) : L_{ij} \in S_b\}$ and $R = \{(i,j) : Y_{ij}^* = 1\}$. Let $\mathcal{P}_\Omega$ be the projection operator on matrices such that
$$(\mathcal{P}_\Omega Z)_{ij} = \begin{cases} Z_{ij} & \text{if } (i,j) \in \Omega, \\ 0 & \text{otherwise.} \end{cases}$$

$\mathcal{P}_R$, $\mathcal{P}_{\Omega \cap R}$ etc are defined similarly.

Let $U\Sigma U^\top$ be the reduced SVD of $Y^*$. Let $\mathcal{C}_i$ be the cluster in which node $i$ belongs and $K_i = |\mathcal{C}_i|$. Note that

$$(UU^\top)_{ij} = \begin{cases} \frac{1}{K_i} & \text{if } \mathcal{C}_i = \mathcal{C}_j \\ 0 & \text{otherwise.} \end{cases} \tag{17}$$

Define the projection operators

$$\mathcal{P}_T Z := UU^\top Z + ZUU^\top - UU^\top ZUU^\top \tag{18}$$

and

$$\mathcal{P}_{T^\perp} Z := Z - \mathcal{P}_T Z.$$

For any matrix $X$ such that $\|X\| \leq \lambda$, $UU^\top + \frac{1}{\lambda}\mathcal{P}_{T^\perp}X$ is a subgradient of $\|\cdot\|_*$ at $Y^*$. For any feasible $Y$, we therefore have

$$\|Y^*\|_* \geq \|Y\|_* \geq \|Y^*\|_* + \langle UU^\top + \frac{1}{\lambda}\mathcal{P}_{T^\perp}X, Y - Y^* \rangle$$

which gives

$$\langle X, Y^* - Y \rangle \geq \langle \mathcal{P}_T X - \lambda UU^\top, Y^* - Y \rangle. \tag{19}$$

We need the following lemmas:

**Lemma 1.** *For any matrix $Z$, we have*

$$(\mathcal{P}_T Z)_{ij} = \frac{1}{K_i} \sum_{k \in \mathcal{C}_i} Z_{kj} + \frac{1}{K_j} \sum_{l \in \mathcal{C}_j} Z_{il} - \frac{1}{K_i K_j} \sum_{k \in \mathcal{C}_i} \sum_{l \in \mathcal{C}_j} Z_{kl}.$$

*Proof.* The lemma is immediate by the definition (18) of $\mathcal{P}_T$ and the expression (17) of $UU^\top$.  □

**Lemma 2.** *With probability at least $1 - n^{-\beta}$ the followings hold:*

$$\|\mathcal{P}_\Omega UU^\top - \mathbb{E}[\mathcal{P}_\Omega UU^\top]\| \leq \lambda_1 \tag{20}$$

*and for all $i, j$*

$$|(\mathcal{P}_T(\mathcal{P}_\Omega UU^\top - \mathbb{E}[\mathcal{P}_\Omega UU^\top]))_{ij}| \leq \begin{cases} \frac{\lambda_1}{K_i} & \text{if } \mathcal{C}_i = \mathcal{C}_j \\ 0 & \text{otherwise} \end{cases} \tag{21}$$

*where*

$$\lambda_1 = c_1 \frac{\beta \log n + \sqrt{K s_\mu (1 - s_\mu) \beta \log n}}{K}.$$

*Proof.* For (20), consider $\mathcal{P}_\Omega UU^\top - \mathbb{E}[\mathcal{P}_\Omega UU^\top]$ as the sum of independent, zero-mean random matrices:

$$\mathcal{P}_\Omega UU^\top - \mathbb{E}[\mathcal{P}_\Omega UU^\top] = \sum_{i < j, \mathcal{C}_i = \mathcal{C}_j} X_{i,j}$$

where

$$X_{i,j} = \mathcal{P}_\Omega \left( \frac{1}{K_i}(e_i e_j^\top + e_j e_i^\top) \right) - \mathbb{E}\mathcal{P}_\Omega \left( \frac{1}{K_i}(e_i e_j^\top + e_j e_i^\top) \right)$$

and $e_i$ denotes the $i$-th vector of the standard basis. Note that

$$\|X_{i,j}\| \leq \frac{1}{K} \quad \forall i, j$$

and

$$\left\| \sum_{i < j, \mathcal{C}_i = \mathcal{C}_j} \mathbb{E}X_{i,j}^2 \right\| = \left\| \sum_{i < j, \mathcal{C}_i = \mathcal{C}_j} \frac{s_\mu(1 - s_\mu)}{K_i^2}(e_i e_i^\top + e_j e_j^\top) \right\| \leq \frac{s_\mu(1 - s_\mu)}{K}$$

By applying matrix Bernstein inequality, we obtain (20).

For (21), let $Z = \mathcal{P}_\Omega UU^\top - \mathbb{E}[\mathcal{P}_\Omega UU^\top]$ be a zero-mean random matrix. For each $i$, note that

$$\hat{Z}_i = \frac{1}{K_i} \sum_{k \in \mathcal{C}_i} Z_{ik}$$

is the average of $K_i$ independent zero-mean random variables with $|Z_{ik}| \leq \frac{1}{K_i}$ and

$$\mathrm{Var}(Z_{ik}) = \frac{s_\mu(1 - s_\mu)}{K_i^2}.$$

By standard Bernstein's inequality, we have that $|\hat{Z}_i| \leq \frac{\lambda_1}{K_i}$ for each $i$. By Lemma 1 and a union bound over all $i$, we obtain (21). □

**Lemma 3.** *With probability at least* $1 - n^{-\beta}$ *the followings hold:*

$$\|\mathcal{P}_{\Omega^c \cap R} W - \mathbb{E}[\mathcal{P}_{\Omega^c \cap R} W]\| \leq \lambda_2 \tag{22}$$

*and for all* $i, j$

$$|(\mathcal{P}_T(\mathcal{P}_{\Omega^c \cap R} W - \mathbb{E}[\mathcal{P}_{\Omega^c \cap R} W]))_{ij}| \leq \begin{cases} \frac{\lambda_2}{K_i} & \text{if } \mathcal{C}_i = \mathcal{C}_j \\ 0 & \text{otherwise} \end{cases} \tag{23}$$

*where*

$$\lambda_2 = c_2 \left( b' \beta \log n + \sqrt{n \mathrm{Var}_\mu(\tilde{w}_b) \beta \log n} \right).$$

*Proof.* It follows the same arguments as in the proof of Lemma 2. □

**Lemma 4.** *With probability at least* $1 - n^{-\beta}$ *the followings hold:*

$$\|\mathcal{P}_{\Omega^c \cap R^c} W - \mathbb{E}[\mathcal{P}_{\Omega^c \cap R^c} W]\| \leq \lambda_3 \tag{24}$$

*and for all* $i, j$

$$|(\mathcal{P}_T(\mathcal{P}_{\Omega^c \cap R^c} W - \mathbb{E}[\mathcal{P}_{\Omega^c \cap R^c} W]))_{ij}| \leq \begin{cases} \lambda_4 & \text{if } \mathcal{C}_i \neq \mathcal{C}_j \\ 0 & \text{otherwise} \end{cases} \tag{25}$$

*where*

$$\lambda_3 = c_3 \left( b' \beta \log n + \sqrt{n \mathrm{Var}_\nu(\tilde{w}_b) \beta \log n} \right)$$

*and*

$$\lambda_4 = c_4 \frac{b' \beta \log n + \sqrt{K \mathrm{Var}_\nu(\tilde{w}_b) \beta \log n}}{K}.$$

*Proof.* It follows the same arguments as in the proof of Lemma 2. □

### B.1.2 Proof of the theorem

We are now ready to complete the proof of Theorem 6. We show that with probability at least $1 - n^{-\beta}$ the following holds for all feasible $Y \neq Y^*$:

$$\langle W, Y^* - Y \rangle > 0$$

which implies that $Y^*$ is the unique solution of program (2).

We decompose $\langle W, Y^* - Y \rangle$ as follows:

$$\langle W, Y^* - Y \rangle = \langle \mathcal{P}_{\Omega \cap R} W, Y^* - Y \rangle + \langle \mathcal{P}_{\Omega \cap R^c} W, Y^* - Y \rangle + \langle \mathcal{P}_{\Omega^c \cap R} W, Y^* - Y \rangle + \langle \mathcal{P}_{\Omega^c \cap R^c} W, Y^* - Y \rangle. \tag{26}$$

By (13), we have that

$$\langle \mathcal{P}_{\Omega \cap R} W, Y^* - Y \rangle \geq 0.$$

It follows that

$$
\begin{aligned}
\langle \mathcal{P}_{\Omega \cap R} W, Y^* - Y \rangle &\geq bK \langle \mathcal{P}_\Omega U U^\top, Y^* - Y \rangle \\
&= bK \left( \langle \mathbb{E}[\mathcal{P}_\Omega U U^\top], Y^* - Y \rangle + \langle \mathcal{P}_\Omega U U^\top - \mathbb{E}[\mathcal{P}_\Omega U U^\top], Y^* - Y \rangle \right) \\
&= bK \left( \langle s_\mu U U^\top, Y^* - Y \rangle + \langle \mathcal{P}_\Omega U U^\top - \mathbb{E}[\mathcal{P}_\Omega U U^\top], Y^* - Y \rangle \right) \\
&\overset{(a)}{\geq} bK \left( \langle s_\mu U U^\top, Y^* - Y \rangle + \langle \mathcal{P}_T(\mathcal{P}_\Omega U U^\top - \mathbb{E}[\mathcal{P}_\Omega U U^\top]) - \lambda_1 U U^\top, Y^* - Y \rangle \right) \\
&\overset{(b)}{\geq} bK \left( \langle s_\mu U U^\top, Y^* - Y \rangle + \langle -2\lambda_1 U U^\top, Y^* - Y \rangle \right) \\
&= bK(s_\mu - 2\lambda_1) \langle U U^\top, Y^* - Y \rangle
\end{aligned}
$$

where we apply (19) and (20) in (a) and (21) in (b). We conclude that

$$\langle \mathcal{P}_{\Omega \cap R} W, Y^* - Y \rangle \geq \max\{0, bK(s_\mu - 2\lambda_1)\} \langle UU^\top, Y^* - Y \rangle \tag{27}$$

Consider the 2nd RHS term of (26). By (13), we have that

$$\langle \mathcal{P}_{\Omega \cap R^c} W, Y^* - Y \rangle \geq 0. \tag{28}$$

Now, consider the 3rd RHS term of (26),

$$
\begin{aligned}
\langle \mathcal{P}_{\Omega^c \cap R} W, Y^* - Y \rangle &= \langle \mathbb{E}[\mathcal{P}_{\Omega^c \cap R} W], Y^* - Y \rangle + \langle \mathcal{P}_{\Omega^c \cap R} W - \mathbb{E}[\mathcal{P}_{\Omega^c \cap R} W], Y^* - Y \rangle \\
&= \langle (\mathbb{E}_\mu \tilde{w}_b) Y^*, Y^* - Y \rangle + \langle \mathcal{P}_{\Omega^c \cap R} W - \mathbb{E}[\mathcal{P}_{\Omega^c \cap R} W], Y^* - Y \rangle \\
&\overset{(a)}{\geq} \langle (\mathbb{E}_\mu \tilde{w}_b) Y^*, Y^* - Y \rangle + \langle \mathcal{P}_T(\mathcal{P}_{\Omega^c \cap R} W - \mathbb{E}[\mathcal{P}_{\Omega^c \cap R} W]) - \lambda_2 UU^\top, Y^* - Y \rangle \\
&\overset{(b)}{\geq} \langle (\mathbb{E}_\mu \tilde{w}_b) Y^*, Y^* - Y \rangle - \langle 2\lambda_2 UU^\top, Y^* - Y \rangle \\
&\geq (K \mathbb{E}_\mu \tilde{w}_b - 2\lambda_2) \langle UU^\top, Y^* - Y \rangle \tag{29}
\end{aligned}
$$

where we use (19) and (22) in (a) and (23) in (b).

Finally, consider the last RHS term of (26),

$$
\begin{aligned}
\langle \mathcal{P}_{\Omega^c \cap R^c} W, Y^* - Y \rangle &= \langle \mathbb{E}[\mathcal{P}_{\Omega^c \cap R^c} W], Y^* - Y \rangle + \langle \mathcal{P}_{\Omega^c \cap R^c} W - \mathbb{E}[\mathcal{P}_{\Omega^c \cap R^c} W], Y^* - Y \rangle \\
&= (-\mathbb{E}_\nu \tilde{w}_b) \langle 1 - Y^*, Y \rangle + \langle \mathcal{P}_{\Omega^c \cap R^c} W - \mathbb{E}[\mathcal{P}_{\Omega^c \cap R^c} W], Y^* - Y \rangle \\
&\overset{(a)}{\geq} (-\mathbb{E}_\nu \tilde{w}_b) \langle 1 - Y^*, Y \rangle + \langle \mathcal{P}_T(\mathcal{P}_{\Omega^c \cap R^c} W - \mathbb{E}[\mathcal{P}_{\Omega^c \cap R^c} W]) - \lambda_3 UU^\top, Y^* - Y \rangle \\
&\overset{(b)}{\geq} (-\mathbb{E}_\nu \tilde{w}_b - \lambda_4) \langle 1 - Y^*, Y \rangle - \langle \lambda_3 UU^\top, Y^* - Y \rangle \tag{30}
\end{aligned}
$$

where we use (19) and (24) in (a) and (25) in (b).

Putting together (27), (28), (29) and (30) we have that

$$
\begin{aligned}
\langle W, Y^* - Y \rangle \geq &(\max\{0, bK(s_\mu - 2\lambda_1)\} + K\mathbb{E}_\mu \tilde{w}_b - 2\lambda_2 - \lambda_3)\langle UU^\top, Y^* - Y \rangle + \\
&(-\mathbb{E}_\nu \tilde{w}_b - \lambda_4)\langle 1 - Y^*, Y \rangle \\
= &bK \left( \max\left(0, s_\mu - 2\lambda_1\right) + \frac{\mathbb{E}_\mu \tilde{w}_b - \frac{2\lambda_2 + \lambda_3}{K}}{b} \right) \langle UU^\top, Y^* - Y \rangle + \\
&(-\mathbb{E}_\nu \tilde{w}_b - \lambda_4)\langle 1 - Y^*, Y \rangle. \tag{31}
\end{aligned}
$$

We see that condition (14) ensures that $-\mathbb{E}_\nu \tilde{w}_b - \lambda_4 \geq 0$. Condition (15) ensures that

$$\mathbb{E}_\mu \tilde{w}_b - \frac{2\lambda_2 + \lambda_3}{K} > 0$$

while condition (16) ensures that

$$s_\mu - 2\lambda_1 + \frac{\mathbb{E}_\mu \tilde{w}_b - \frac{2\lambda_2 + \lambda_3}{K}}{b} > 0$$

and either one is sufficient.

## C  Proofs of Theorems 1–2 and Corollary 1

In this section we prove the main theoretical results in the main text: Theorems 1 for using arbitrary weights, and Theorem 2 and Corollary 1 for using MLE weights.

### C.1  Proof of Theorem 1

Theorem 1 is a special case of Theorem 6 by setting $b$ to $\infty$ and $\beta = 10$.

## C.2 Proof of Theorem 2

Throughout the remainder of this section, $w$ always means $w^{\text{MLE}}$. The proof is done using several lemmas. The first lemma, proved in subsection C.2.1 to follow, bounds the log-likelihood ratio.

**Lemma 5.** *Suppose that* $\left|\log \frac{\mu}{\nu}\right| \leq b$. *Then, for any* $l \in \mathcal{L}$,

$$\left|\log \frac{\mu(l)}{\nu(l)}\right| \leq (b+2) \left|\frac{\mu(l) - \nu(l)}{\mu(l) + \nu(l)}\right|.$$

The second lemma, proved in section C.2.2 to follow, controls the variance terms.

**Lemma 6.** *Suppose that* $\left|\log \frac{\mu}{\nu}\right| \leq b$ *and* $D(\nu||\mu) \leq \zeta D(\mu||\nu)$, *then*

$$\text{Var}_\nu w \leq 3(b+2)D(\nu||\mu) \tag{32}$$

*and*

$$\max(\text{Var}_\mu w, \text{Var}_\nu w) \leq (\zeta + 1)(b + 2)D(\mu||\nu). \tag{33}$$

The last lemma is a classical result in information theory that bounds the KL divergences $D(\mu||\nu)$ and $D(\nu||\mu)$ with the triangle discrimination between $\mu$ and $\nu$.

**Lemma 7** ([24]). *The following holds for any distributions $\mu$ and $\nu$:*

$$\min\{D(\mu||\nu), D(\nu||\mu)\} \geq \frac{1}{2} \sum_{l \in \mathcal{L}} \frac{(\mu(l) - \nu(l))^2}{\mu(l) + \nu(l)}.$$

We are now ready to prove Theorem 2. To this end we shall verify that the conditions (3) and (4) in Theorem 1 holds under the assumption of Theorem 2. Note that

$$-\mathbb{E}_\nu w = -\sum_l \nu(l) \log \frac{\mu(l)}{\nu(l)} = D(\nu||\mu).$$

Combining (5) with (32) yields (3). Turning to (4), we note that $\mathbb{E}_\mu w = D(\mu||\nu)$. Combining (6) with (33) gives (4). Finally, the last sentence in the statement of Theorem 2 is immediate from the following more general result, which is useful later in the proof of Theorem 3.

**Lemma 8.** *Suppose that* $\left|\log \frac{\mu}{\nu}\right| \leq b$, *then we have*

$$D(\nu||\mu) + D(\mu||\nu) \leq (b+2) \sum_l \frac{(\nu(l) - \mu(l))^2}{\nu(l) + \mu(l)}$$

*and*

$$\frac{1}{2b+3} \leq \frac{D(\nu||\mu)}{D(\mu||\nu)} \leq 2b + 3.$$

*Proof.* We have

$$\begin{aligned}
\frac{D(\nu||\mu)}{D(\mu||\nu)} + 1 &= \frac{D(\nu||\mu) + D(\mu||\nu)}{D(\mu||\nu)} \\
&= \frac{\sum_l (\nu(l) - \mu(l)) \log \frac{\nu(l)}{\mu(l)}}{D(\mu||\nu)} \\
&\overset{(a)}{\leq} \frac{(b+2) \sum_l \frac{(\nu(l) - \mu(l))^2}{\nu(l) + \mu(l)}}{D(\mu||\nu)}.
\end{aligned}$$

where we apply Lemma 5 in (a). This proves the first equation in the lemma. Bounding the last RHS using Lemma 7, we prove the second inequality in the second equation in the lemma. The first inequality in the second equation follows from switching the roles of $\mu$ and $\nu$. $\qquad\square$

### C.2.1 Proof of Lemma 5

*Proof.* Consider the function $\log \frac{1-p}{p}$ for $p \in [\frac{1}{e^b+1}, 0.5]$. By its convexity in this range we can linearly upper-bound it and show that $\log \frac{1-p}{p} \leq b\left(\frac{e^b+1}{e^b-1}\right)(1-2p)$. Now take $p = \frac{\nu}{\mu+\nu}$, we then have

$$\left|\log \frac{\mu}{\nu}\right| = \left|\log \frac{1-p}{p}\right| \leq b\left(\frac{e^b+1}{e^b-1}\right)|1-2p| = b\left(\frac{e^b+1}{e^b-1}\right)\left|\frac{\mu-\nu}{\mu+\nu}\right| \leq (b+2)\left|\frac{\mu-\nu}{\mu+\nu}\right|.$$

$\square$

### C.2.2 Proof of Lemma 6

*Proof.* Using Lemma 5, we have

$$\mathrm{Var}_\nu w \leq \mathbb{E}_\nu w^2$$
$$= \sum_l \nu(l)\left(\log \frac{\mu(l)}{\nu(l)}\right)^2$$
$$\leq (b+2)\sum_l \nu(l)\left|\frac{\mu(l)-\nu(l)}{\mu(l)+\nu(l)}\right|\left|\log \frac{\mu(l)}{\nu(l)}\right|$$
$$\overset{(a)}{\leq} 3(b+2)\sum_l \nu(l)\log \frac{\nu(l)}{\mu(l)}$$
$$= 3(b+2)D(\nu\|\mu).$$

For (a), we have

$$3\sum_l \nu(l)\log \frac{\nu(l)}{\mu(l)} - \sum_l \nu(l)\left|\frac{\mu(l)-\nu(l)}{\mu(l)+\nu(l)}\right|\left|\log \frac{\mu(l)}{\nu(l)}\right| = 2\sum_l \nu(l)\frac{2\mu(l)+\nu(l)}{\mu(l)+\nu(l)}\log \frac{\nu(l)}{\mu(l)}$$
$$\overset{(b)}{\geq} 2\sum_l \nu(l)\frac{2\mu(l)+\nu(l)}{\mu(l)+\nu(l)}\frac{(1-\frac{\mu(l)}{\nu(l)})(5+\frac{\mu(l)}{\nu(l)})}{2+4\frac{\mu(l)}{\nu(l)}}$$
$$= \sum_l \frac{(\nu(l)-\mu(l))(5\nu(l)+\mu(l))}{(\mu(l)+\nu(l))}$$
$$= -4 + 8\sum_l \frac{\mu(l)^2}{\mu(l)+\nu(l)}$$
$$\overset{(c)}{\geq} 0$$

where we use Pade's bound for logarithm in (b). In (c) note that $\sum_l \frac{\mu(l)^2}{\mu(l)+\nu(l)} \geq \frac{1}{2}$.

For (33), again by using Lemma 5,

$$\max(\mathrm{Var}_\mu w, \mathrm{Var}_\nu w) \leq \mathbb{E}_\mu w^2 + \mathbb{E}_\nu w^2$$
$$= \sum_l (\mu(l)+\nu(l))\left(\log \frac{\mu(l)}{\nu(l)}\right)^2$$
$$\leq (b+2)\sum_l (\mu(l)+\nu(l))\left|\frac{\mu(l)-\nu(l)}{\mu(l)+\nu(l)}\right|\left|\log \frac{\mu(l)}{\nu(l)}\right|$$
$$= (b+2)(D(\mu\|\nu)+D(\nu\|\mu))$$
$$\leq (\zeta+1)(b+2)D(\mu\|\nu).$$

$\square$

### C.3 Proof of Corollary 1

The corollary follows immediately from Theorem 2 by lower bounding the LHS of (5) and (6) using Lemma 7.

## D Proof of Theorem 3

In this section we prove the converse result in Theorem 3. We use a standard technique based on converting the problem to multiple hypothesis testing; in particular, we shall apply Theorem 2.5 of [27]. Set $M = n$ and let $\theta_0 = Y^*$. For $k = 1, \ldots, \frac{M}{2}$, let $\theta_k$ be the adjacency matrix of a new clustering by swapping the 1st member of cluster 1 with the $k$-th member of cluster 2. Likewise, for $k = \frac{M}{2} + 1, \ldots, M$, $\theta_k$ is obtained by swapping the 2nd member of cluster 1 with the $k$-th member of cluster 2. Let $L^0, L^1, \ldots, L^M$ be the random label matrices generated by the corresponding clustering.

Since the label of each entry is generated independently, we have:

$$
\begin{aligned}
D(L^j || L^0) &= \sum_{i<j} D(L^k_{ij} || L^0_{ij}) \\
&\overset{(a)}{=} (n-2)D(\mu || \nu) + (n-2)D(\nu || \mu) \\
&\overset{(b)}{\leq} (n-2)(b+2) \sum_l \frac{(\nu(l) - \mu(l))^2}{\nu(l) + \mu(l)} \\
&\overset{(c)}{\leq} (c'+2)c \log n;
\end{aligned}
$$

here in (a) we use the fact that due to the membership swap, exactly $n - 2$ intra-cluster pairs in $\theta_0$ become inter-cluster pairs in $\theta_j$ and vise-versa, (b) follows from the first equation in Lemma 8, and (c) holds due to (8) and the assumption that $b$ is bounded by a universal constant.

The result then follows from taking $c$ sufficiently small and applying Theorem 2.5 of [27].

## E Proof of Theorem 4

In this section we prove the monotonicity property in Theorem 4. Let $E = \{l \in \mathcal{L} : \bar{\mu}(l) \geq \bar{\nu}(l)\}$ and $E^c = \mathcal{L} \setminus E$. Since $(\mu, \nu)$ is strictly more divergent than $(\bar{\mu}, \bar{\nu})$, we have that for all $l \in E$, $\mu(l) \geq \bar{\mu}(l) \geq \bar{\nu}(l) \geq \nu(l)$ and for all $l \in E^c$, $\nu(l) \geq \bar{\nu}(l) > \bar{\mu}(l) \geq \mu(l)$.

We generate the label matrix $L$ from $(\mu, \nu)$ using the following two-stage process:

1. First, generate a matrix $\bar{L}$ from $(\bar{\mu}, \bar{\nu})$. Set $L \leftarrow \bar{L}$.

2. • For each $(i, j)$ where $Y^*_{ij} = 0$, if $L_{ij} \in E$, then with probability $1 - \frac{\nu(L_{ij})}{\bar{\nu}(L_{ij})}$, set $L_{ij} \leftarrow l$ where $l$ is drawn from the set $E^c$ with distribution $\frac{\nu(l) - \bar{\nu}(l)}{\sum_{l' \in E^c} \nu(l') - \bar{\nu}(l')}$. Let $\Omega_-$ be the set of all such entries, i.e. where $L_{ij}$ has switched from $E$ to $E^c$.

   • For each $(i, j)$ where $Y^*_{ij} = 1$, if $L_{ij} \in E^c$, then with probability $1 - \frac{\mu(L_{ij})}{\bar{\mu}(L_{ij})}$, set $L_{ij} \leftarrow l$ where $l$ is drawn from the set $E$ with distribution $\frac{\mu(l) - \bar{\mu}(l)}{\sum_{l' \in E} \mu(l') - \bar{\mu}(l')}$. Let $\Omega_+$ be the set of all such entries, i.e. where $L_{ij}$ has switched from $E^c$ to $E$.

It is straightforward to verify that the resulting distribution of $L$ is identical to that generated by $(\mu, \nu)$.

Let $\widehat{Y}$ be the optimal solution to (2) with $\bar{L}$ as input. Let $\bar{W}$ be the corresponding MLE weights. Since $(\bar{\mu}, \bar{\nu})$ satisfies the condition of Theorem 2, we have that with probability at least $1 - n^{-\beta}$, $\widehat{Y} = Y^*$ and for any feasible solution $Y \neq Y^*$, we have

$$
\langle \bar{W}, Y^* \rangle > \langle \bar{W}, Y \rangle.
$$

Now, consider program (2) with $L$ as the input, using the corresponding MLE weights $W$ based on $(\bar{\mu}, \bar{\nu})$. We have that for any feasible $Y \neq Y^*$,

$$
\begin{aligned}
\langle W, Y^* \rangle - \langle \bar{W}, Y^* \rangle &= \sum_{(i,j) \in \Omega_+} (W_{ij} - \bar{W}_{ij}) Y_{ij}^* \\
&\geq \sum_{(i,j) \in \Omega_+} (W_{ij} - \bar{W}_{ij}) Y_{ij} \\
&\geq \sum_{(i,j) \in \Omega_-} (W_{ij} - \bar{W}_{ij}) Y_{ij} + \sum_{(i,j) \in \Omega_+} (W_{ij} - \bar{W}_{ij}) Y_{ij} \\
&= \langle W, Y \rangle - \langle \bar{W}, Y \rangle \\
\Rightarrow \quad \langle W, Y^* \rangle - \langle W, Y \rangle &\geq \langle \bar{W}, Y^* \rangle - \langle \bar{W}, Y \rangle > 0.
\end{aligned}
$$

# F    Proof of Theorem 5

In this section we prove Theorem 5, which provides guarantees for using inaccurate weights. We apply Theorem 1. To avoid confusion we use $b'$ and $c'$ to denote the constants $b$ and $c$ in Theorem 1.

For (3), from the RHS, we have

$$
\begin{aligned}
c' \frac{b' \log n + \sqrt{K \log n} \sqrt{\mathrm{Var}_\nu w}}{K} &\overset{(a)}{\leq} \frac{c' b' (1-\gamma)^2}{c \lambda^2 (b+2)} D(\nu \| \mu) + \sqrt{3 c'^2 \lambda^2 (b+2) D(\nu \| \mu) \frac{\log n}{K}} \\
&\leq \frac{c'(1-\gamma)^2}{c \lambda} D(\nu \| \mu) + \sqrt{\frac{3 c'^2 (1-\gamma)^2}{c} D(\nu \| \mu)^2} \\
&\overset{(b)}{\leq} (1-\gamma) D(\nu \| \mu) \\
&\overset{(c)}{\leq} D(\nu \| \mu) - \Delta_\nu \\
&= -\mathbb{E}_\nu w
\end{aligned}
$$

where for (a), we use $\mathrm{Var}_\nu w \leq \lambda^2 \mathrm{Var}_\nu w^{\mathrm{MLE}}$ due to the condition $|w| \leq \lambda |\log \frac{\mu}{\nu}|$ and apply (32). For (b) we choose an appropriately large $c$. For (c) we use the condition $|\Delta_\nu| \leq \gamma D(\nu \| \mu)$.

The same arguments can be used to prove (4) where (33) is used instead of (32).

# G    Proof of Corollary 2

In this section we prove Corollary 2 for clustering Gaussian graphs. The only difficulty in applying Theorem 2 directly is the boundedness condition $|w^{\mathrm{MLE}}(L_{ij})| \leq b$. To overcome this we use a standard truncation trick. Define a truncated version $\bar{L}$ of the matrix $L$ by $\bar{L}_{ij} = L_{ij} \cdot \mathbf{1}_{\{L_{ij} = ? \text{ or } |L_{ij} - u_{ij}| \leq c' \sqrt{\log n}\}}$, where $\mathbf{1}$ is the indicator function and $c' > 0$ is a universal constant that is sufficiently large. By a standard tail bound on Gaussian variables and the union bound, we know that with probability at least $1 - n^{-10}$, $\bar{L}_{ij} = L_{ij}$ for all $i, j$. Therefore, any result for $\bar{L}$ also holds for $L$ with that probability. Now with $\bar{L}$ we have a new labeled problem defined on $\mathcal{L} = [\underline{u} - c' \sqrt{\log n}, \bar{u} + c' \sqrt{\log n}] \cup \{?\}$. We use the weight $w(\bar{L}_{ij}) = \bar{L}_{ij} - \frac{\bar{u} + \underline{u}}{2}$ if $\bar{L}_{ij} \neq ?$ and $w(\bar{L}_{ij}) = 0$ otherwise, which is bounded almost surely. Applying Theorem 1 gives the desired result.

# H   Proof of Corollary 5

In this section we prove Corollary 5 for clustering with Markov snapshots. The MLE weight in this case is given by:

$$w^{\text{MLE}}(\bar{L}_{ij}) = \frac{1}{T} \log \frac{\bar{\mu}(L_{ij}^{(1)}) \mu(L_{ij}^{(2)}|L_{ij}^{(1)}) \dots \mu(L_{ij}^{(T)}|L_{ij}^{(T-1)})}{\bar{\nu}(L_{ij}^{(1)}) \nu(L_{ij}^{(2)}|L_{ij}^{(1)}) \dots \nu(L_{ij}^{(T)}|L_{ij}^{(T-1)})}$$

$$= \frac{1}{T} \log \frac{\bar{\mu}(L_{ij}^{(1)})}{\bar{\nu}(L_{ij}^{(1)})} + \frac{1}{T} \sum_{t=2}^{T} \log \frac{\mu(L_{ij}^{(t)}|L_{ij}^{(t-1)})}{\nu(L_{ij}^{(t)}|L_{ij}^{(t-1)})}.$$

In the sequel, we will focus on an in-cluster pair $i, j$ with label distribution $\mu$ and drop the subscript $ij$ in $L_{ij}$. The same analysis holds for all $i, j$.

It is convenient to think of an auxiliary Markov chain $X_1, \dots X_T$ where each state is characterized by a label pair $X_t = (L^{(t-1)}, L^{(t)})$ for $t > 1$, and $X_1 = L^{(1)}$. We define the function $f$ on $X_t$ such that

$$f(L) = \log \frac{\bar{\mu}(L)}{\bar{\nu}(L)} \quad \text{and} \quad f(L, L') = \log \frac{\mu(L'|L)}{\nu(L'|L)}.$$

We therefore have

$$w^{\text{MLE}} = \frac{1}{T} \sum_{t=1}^{T} f(X_t).$$

It is straightforward to show that

$$\mathbb{E}_\mu(f(X_1)) = D(\bar{\mu}||\bar{\nu}) \quad \text{and} \quad \mathbb{E}_\mu(f(X_t)) = \mathbb{E}_{\bar{\mu}} D_l(\mu||\nu) \quad (t > 1)$$

therefore

$$\mathbb{E}_\mu(w^{\text{MLE}}) = \frac{1}{T} D(\bar{\mu}||\bar{\nu}) + \left(1 - \frac{1}{T}\right) \mathbb{E}_{\bar{\mu}} D_l(\mu||\nu). \tag{34}$$

The rest of the proof concerns with bounding $\text{Var}_\mu(w^{\text{MLE}})$. Following the proof of Lemma 6, the variance of $f(X_t)$ can be bounded by

$$\text{Var}_\mu(f(X_1)) \leq 3(b+2)D(\bar{\mu}||\bar{\nu}) \quad \text{and} \quad \text{Var}_\mu(f(X_t)) \leq 3(b+2)\mathbb{E}_{\bar{\mu}} D_l(\mu||\nu) \quad (t > 1).$$

We now bound the covariance $\text{Cov}(f(X_t), f(X_{t+\tau+1}))$ for $t \geq 2, \tau \geq 0$,

$$\text{Cov}(f(X_t), f(X_{t+\tau+1}))$$

$$= \mathbb{E}_\mu \left[ \log \frac{\mu(L^{(t)}|L^{(t-1)})}{\nu(L^{(t)}|L^{(t-1)})} \log \frac{\mu(L^{(t+\tau+1)}|L^{(t+\tau)})}{\nu(L^{(t+\tau+1)}|L^{(t+\tau)})} \right] - \mathbb{E}_{\bar{\mu}} D_l(\mu||\nu)^2$$

$$= \sum_{L^{(t-1)}} \bar{\mu}(L^{(t-1)}) \sum_{L^{(t)}} \mu(L^{(t)}|L^{(t-1)}) \log \frac{\mu(L^{(t)}|L^{(t-1)})}{\nu(L^{(t)}|L^{(t-1)})} \sum_{L^{(t+\tau)}} \Pr(L^{(t+\tau)}|L^{(t)}) D_{L^{(t+\tau)}}(\mu||\nu)$$

$$\quad - \mathbb{E}_{\bar{\mu}} D_l(\mu||\nu)^2$$

$$\overset{(a)}{\leq} \sum_{L^{(t-1)}} \bar{\mu}(L^{(t-1)}) \sum_{L^{(t)}} \mu(L^{(t)}|L^{(t-1)}) \left| \log \frac{\mu(L^{(t)}|L^{(t-1)})}{\nu(L^{(t)}|L^{(t-1)})} \right| \sum_{L^{(t+\tau)}} \kappa \gamma^\tau D_{L^{(t+\tau)}}(\mu||\nu)$$

$$\leq \frac{\kappa \gamma^\tau}{\min_l \bar{\mu}(l)} b \mathbb{E}_{\bar{\mu}} D_l(\mu||\nu)$$

where in (a) we use the geometric ergodicity of $\mu$ in the sense that

$$|\Pr(L^{(t+\tau)}|L^{(t)}) - \bar{\mu}(L^{(t+\tau)})| \leq \kappa \gamma^\tau.$$

The same bound also applies to the case $t = 1$. Note that the covariance bound is independent of $t$ and only dependent on $\tau$.

We proceed to bound $\text{Var}(w^{\text{MLE}})$ as follows

$$
\begin{aligned}
\text{Var}(w^{\text{MLE}}) &= \frac{1}{T^2}\sum_{t=1}^{T}\text{Var}(f(X_t)) + \frac{2}{T^2}\sum_{t=1}^{T-1}\sum_{t'=t+1}^{T}\text{Cov}(f(X_t),f(X_{t'})) \\
&\leq \frac{1}{T^2}\sum_{t=1}^{T}\text{Var}(f(X_t)) + \frac{2}{T}\sum_{\tau=0}^{T-2}\frac{T-1-\tau}{T}\frac{\kappa\gamma^\tau}{\min_l\bar{\mu}(l)}b\mathbb{E}_{\bar{\mu}}D_l(\mu||\nu) \\
&\leq \frac{3(b+2)}{T}\left(\frac{1}{T}D(\bar{\mu}||\bar{\nu}) + \frac{T-1}{T}\mathbb{E}_{\bar{\mu}}D_l(\mu||\nu)\right) + \frac{2\kappa}{(1-\gamma)\min_l\bar{\mu}(l)}b\frac{\mathbb{E}_{\bar{\mu}}D_l(\mu||\nu)}{T} \\
&\leq c\frac{(b+2)\lambda}{T}\left(\frac{1}{T}D(\bar{\mu}||\bar{\nu}) + \frac{T-1}{T}\mathbb{E}_{\bar{\mu}}D_l(\mu||\nu)\right) \quad (35)
\end{aligned}
$$

With (34) and (35) we can now finish the proof by applying Theorem 1.

# I  Example Markov chain with explicit bound on $\lambda$

The snapshots in a Markov label sequence are not independent and therefore given $T$ snapshots we do not expect a $T$-fold increase in the information as in the independent case. In the condition given by Corollary 5, this penalty is characterized by an extra constant $\lambda$. To give a sense of what values it may take, we now derive an explicit bound for a simple class of 2-label sequences.

Suppose the transition matrix for $\mu$ is given by the following for $0 < p_0, p_1 < 1$:

$$
\begin{bmatrix} 1-p_0 & p_0 \\ p_1 & 1-p_1 \end{bmatrix}.
$$

We can assume that $p_0$ is the probability that an edge "flips" into a non-edge and $p_1$ is the probability that a non-edge flips into an edge.

Let $\rho = 1-p_0-p_1$. By eigen-decomposition we can show that the transition matrix after $t$ transitions is:

$$
\begin{bmatrix} \frac{p_1}{p_0+p_1}\left(1+\frac{p_0}{p_1}\rho^t\right) & \frac{p_0}{p_0+p_1}(1-\rho^t) \\ \frac{p_1}{p_0+p_1}(1-\rho^t) & \frac{p_0}{p_0+p_1}\left(1+\frac{p_1}{p_0}\rho^t\right) \end{bmatrix}
$$

and the stationary distribution is given by

$$
\bar{\mu} = \begin{bmatrix} \dfrac{p_1}{p_0+p_1} & \dfrac{p_0}{p_0+p_1} \end{bmatrix}.
$$

It is easy to see that this satisfies the geometric ergodicity condition

$$
|\Pr(L^{(1+t)}|L^{(1)}) - \bar{\mu}(L^{(1+t)})| \leq \kappa\gamma^t
$$

with $\kappa = 1$ and $\gamma = |\rho|$. We therefore have

$$
\lambda = \frac{p_0+p_1}{(1-|\rho|)\min\{p_0,p_1\}}.
$$

For fixed $\bar{\mu}$, the value of $\lambda$ determines how much new information is contained in a new snapshot. For example, suppose $p_1 = p_0$, so $\bar{\mu} = [\frac{1}{2},\frac{1}{2}]$ is fixed. The value of $\lambda$ is very large when the flipping probabilities $p_0$ and $p_1$ are both close to zero (or close to one). In this case the next snapshot is almost either the same or the inverse of the current snapshot, hence providing very little extra information. In the other extreme, if $p_0 = p_1 = \frac{1}{2}$, then $\lambda = 2$ is small. In this case the snapshots are independent and thus the next snapshot provides fresh information.