[Reviews · NeurIPS 2014]

Submitted by Assigned_Reviewer_6

+++++++++
Summary of comments:

Pros:
- Clear and potentially impactful contributions.
- Natural SDP-like formulation and fairly tight subsequent analysis (with matching lower bounds in some cases).
- The general theorem subsumes a number of specific existing results for different probabilistic graph models.
- Extension to time-varying graphs with Markov dependencies is straightforward and elegant.

Cons:
- Some amount of discussion on proof techniques / open questions in the main text would be useful.

++++++++++
Detailed comments:

The paper proposes a general approach for tackling graph clustering problems. In the model considered, every pair of nodes is associated with a ``label" (an element of an arbitrary set), and the authors' approach attempts to discover a clustering that maximizes a certain weighted payoff constructed from the observed labels. The authors relax this combinatorial problem into a natural SDP, and derive sufficient conditions for which the SDP recovers the true clusters. The authors also show that a log-likelihood ratio weight function achieves nearly optimal results. For some well-known probabilistic graph models (e.g., Gaussian graphs, planted partitions, etc), this approach subsumes and generalizes existing results. Since the notion of a ``label” is very generic, the framework also motivates new theoretical results in the context of time-varying graphs (where the graph snapshots are modeled as a Markov chain). Numerical experiments on synthetic and real data support the utility of the technique.

The paper is well written, and the contributions are clear and potentially impactful. The main theorem is very general (and the authors have provided a seemingly stronger result in the Appendix which can handle unbounded weight functions). I also like that the authors robustly characterize the case when imperfect weight functions are used.

Comments/questions: (i) Some intuition/discussion on the techniques used in the analysis would be nice. As of now, the paper does not include proofs and reads more like a compendium of results. (ii) The authors could comment on whether enforcing symmetry/PSD-ness of Y can achieve sharper bounds than (3) or (4). (iii) What are the potential barriers to proving a lower bound for arbitrary number of clusters? (iv) Figure 3: clarify why the performance of the Markov model appears to slightly dip beyond f = 0.005.
Summary: A very well-written paper that develops a general framework for graph clustering. Specific realizations of the framework reproduce a number of existing results, and some novel extensions lead to results for more interesting graph classes (including time-varying graphs.)

Submitted by Assigned_Reviewer_45

This paper proposes a convex-optimization-based framework for graph clustering that provides theoretical guarantees that generalize and improve on previously existing theoretical results on graph clustering. This construction permits applications beyond traditional static graph clustering problems.

I am not an expert in this area, so I am not able to confirm the novelty of the work, but it seems as though the theoretical framework is quite powerful and yields some nice results. The O(n^2) scaling of the algorithm limits its applicability to large graphs, but there are plenty of important applications involving smaller graphs. The empirical results are fine for a paper like this that is focused on theory.

The paper is clear and well written. One minor suggestion regarding presentation: seeing n^{-10} in theorem 1 is a little jarring! The supplement makes it clear that the choice of 10 is arbitrary; it would be good to say this explicitly in the main text.
Summary: This is a nice paper that proposes a graph clustering framework with strong (and general) theoretical guarantees.

Submitted by Assigned_Reviewer_46

Clustering from Labels and Time-Varying Graphs
The authors consider graph clustering i.e. nodes are assigned to groups based on dense interconnectivity, with no/sparse connectivity to other nodes not in the same group. As pointed out by the authors, their model is in the vein of the stochastic labelled block model. However, they can cover additional scenarios such as time-varying graphs.
The basic scenario is presented in eqn (1). This is restated as a relaxed convex optimisation problem in (2) (incidentally a bug ‘and thus and has nuclear …’). The only concern with (2) is that, though convex, and a linear objective, it may be computational intensive for large graphs. The experiment with n=400 in the Supplementary suggests otherwise but it would have been interesting to know algorithmic time complexity dependence on n.
The main theorems in the text give bounds on the estimate of the true cluster matrix Y for the convex program (2). Specifically, Theorem 1 gives sufficient conditions for this convex program to find the true cluster matrix, dependent on label distribution, minimum size of the true clusters (note: I don’t think you would know this in reality?) and the weight function. I wasn’t sure of the significance of c (a universal constant) here. Thus the later Corollary 3 gives a condition on recovering Y which has some support in the experimental Section 5 (below Fig. 1-3), for a particular special case. I was only able to check two lemmas in the Supplementary – though argument looked sound. In section 4 the author(s) move from the general argument of Section 3 to two special cases (e.g. non-uniform edge densities, Corollary 3). This includes Markov sequence and time-varying e.g. Corollary 5 presents sufficient conditions on finding the true cluster matrix based on the same dependency on c, \mu, \nu, K and n for the case of Markov snapshots.
Section 5 presents some brief experimental results (there is no Conclusion), though the Supplementary (Section A) gives more detail about the experiments.
In short, the authors present a convex program for graph clustering which has broader implantation to time-varying graphs. They present criteria for establishing if the true cluster matrix has been found – which appears sound and has experimental support (see e.g. Figure 4 in the Supplementary). The criterion (Theorem 1, for the general case) appears useable (possibly as a stopping criterion – or if stopped early, whether a sound solution has been achieved).
I think the topic of graph clustering is important in terms of real-word applications in bioinformatics, social networks, etc, and this paper looks like a valid and useful contribution. Thus I recommend acceptance of the paper.

Summary: The authord propose a graph clustering method with additional theoretical support for when a suitable cluster matrix has been achieved. The method is baased on a convex program.
Author Feedback
Author rebuttal: We thank all the reviewers for providing very helpful comments.

1. To Assigned_Reviewer_45:

Indeed the value of the exponent in n^{-10} can be any fixed constant. We will clarify this in the main text.

2. To Assigned_Reviewer_46:

Regarding computational complexity, while a generic interior point solver for the semidefinite program can be slow, a more problem-specific solver can handle moderately large n. The Alternating Direction Method of Multipliers (ADMM) solver we used (based on Lin et al's) takes about 1 minute to solve an instance with n=1000 on a typical PC. Developing fast algorithms for solving convex programs, including for the application of clustering, is no doubt an area of ongoing research.

We also note that while the theoretical guarantees depend on the minimum cluster size K, the algorithm itself does not need to know K.

3. To Assigned_Reviewer_6:

(i) Discussion on proof techniques in the analysis: Indeed a good suggestion. We will revise accordingly.

(ii) Using the symmetry/PSD-ness constraints is a possibility that we have not focused on. All our theoretical guarantees carry over with these constraints, but we are not able to prove a sharper bound (our current bound is already tight in certain cases). There have been work on graph clustering (e.g. Mathieu 2010) based on this and it is an interesting direction to further explore these alternative formulations.

(iii) Our proof of the lower bound can be extended to arbitrary number of clusters in principle, with the constant c depending potentially on the number of clusters.

(iv) Repeated runs of the experiment show that the performance graph pretty much levels off at around 90% accuracy after f=0.004. The dip in Fig. 3 is most likely just noise - each repeated run shows a slightly different pattern (rise/dip at different points) where all error bars still overlap.